# Reef visitors' observation of assisted coral recovery devices *in situ* reduces concern about their use

Matthew I. Curnock[1]*, Rhea Arya[1], Emilee Chamberland[1], Katherine Chartrand[2], John Edmondson[3], Eric E. Fisher[4], Rebecca Forster[5], Stewart Lockie[6], Jennifer Loder[7], Danielle Nembhard[6], Abigail Scott[2], Bruce Taylor[8], Jasmina Uusitalo[2]

1 CSIRO Environment, Australian Tropical Science and Innovation Precinct, James Cook University, Townsville, Queensland, Australia, 2 TropWATER, James Cook University, Smithfield, Cairns, Queensland, Australia, 3 Wavelength Reef Cruises, Port Douglas, Queensland, Australia, 4 GBR Biology, Cairns, Queensland, Australia, 5 Australian Institute of Marine Science, Townsville, Queensland, Australia, 6 The Cairns Institute, James Cook University, Smithfield, Cairns, Queensland, Australia, 7 Great Barrier Reef Foundation, South Brisbane, Queensland, Australia, 8 CSIRO Environment, Dutton Park, Brisbane, Queensland, Australia

* matt.curnock@csiro.au

**Data Availability Statement:** The data that support the findings of this study are publicly available from the CSIRO online Data Access Portal (https://doi.org/10.25919/7353-m763).

## Abstract

Assisted coral recovery (ACR) initiatives are establishing rapidly in coral reefs worldwide, using a variety of devices and techniques. In the Great Barrier Reef (GBR, the Reef), site-scale ACR field trials are occurring at multiple sites in the Cairns-Port Douglas region through Reef stewardship activities involving GBR tourism operators, Traditional Owners, and not-for-profit organisations. It is hypothesised that these field trials and the presence of ACR devices at reef tourism sites do not negatively affect visitor experiences, and when accompanied by appropriate educational information, can potentially help to raise awareness of Reef stewardship and conservation efforts. We tested these hypotheses using a survey of 708 Reef visitors on five tourism vessels, 346 of whom reported observing ACR devices *in situ* during their coral reef experience. Ordinal regression tests of survey responses found no statistical relationship between respondents' observation of ACR devices and (i) their overall Reef trip satisfaction, (ii) the perceived aesthetic beauty of the site(s) they visited, and (iii) their concern about the future health of the GBR. However, Reef visitors who observed ACR devices showed significantly lower levels of concern about the use of these devices on the Reef. The perceived quality of educational information presented to respondents was among the significant factors associated with their reef trip satisfaction and perceived beauty of reef sites. Our findings have implications for ACR practitioners and proponents who are concerned about public visibility, perceptions, and support for ACR initiatives, as the scale of such initiatives is expected to increase.

**Funding:** This study was conducted as part of research activities by the Reef Restoration and Adaptation Program's (RRAP) Best Practice Stakeholder and Traditional Owner Engagement Subprogram (https://gbrrestoration.org/program/engagement/), with funding provided by the partnership between the Australian Government's Reef Trust and the Great Barrier Reef Foundation, delivered in partnership with CSIRO, the Great Barrier Reef Marine Park Authority, and the Queensland Government's Reef Water Quality Program. We note that one co-author is an employee of one of the study's funding organisations (J. Loder; Great Barrier Reef Foundation), who contributed to the study's conceptual design and preparation of the manuscript. This co-author was not directly associated with the project's funding or its administration and was not involved in the data collection and analysis, nor in the decision to publish the article.

**Competing interests:** The authors have declared that no competing interests exist.

## Introduction

It has been estimated that more than 430 million people worldwide depend on coral reefs to some extent for their provisioning, regulating and cultural ecosystem services [1]. However, coral reef ecosystems are threatened by multiple anthropogenic pressures that are impacting their values and resilience. Among these pressures, recurrent marine heatwaves driven by climate change are causing coral bleaching and mortality events of increasing scale and severity, which threaten to fundamentally alter coral reefs' ecology, function, and values [2–4].

Traditional approaches to coral reef protection, encompassing the management of direct, localised pressures (e.g., fisheries, land-based runoff, coral predators), are widely recognised as important but insufficient for preserving coral reefs as oceans warm rapidly [5–7]. Consequently, new technological approaches are being developed and tested to help restore degraded reefs at increasing scales, and to improve the tolerance of corals to warmer waters [8, 9].

Assisted coral recovery (ACR; often referred to as *coral restoration*) aims to accelerate the replenishment of corals at damaged or degraded reefs and increase reef resilience. ACR encompasses a variety of techniques which foster the growth of juvenile corals or fragments on devices at reefs or in aquaria prior to colonies being deployed or 'out-planted' back onto the reef substrate, as well as techniques to stabilise rubble habitats and thereby allow coral regrowth, and techniques to enhance larval settlement onto reef substrate [9–11]. Globally, over the past two decades, ACR projects have been labour-intensive and applied at reef sites rarely exceeding one hectare [10]. However, ACR techniques are advancing rapidly, incorporating new science to breed heat tolerant corals, and engineering to increase the scale of production and spatial extent of future deployments [9, 12–14].

### Public perceptions and support for assisted coral recovery

Public perceptions and public acceptance of environmental protection interventions are critical, underpinning political support and public resourcing [15–17]. Whether stated explicitly or otherwise, most ACR initiatives aim to restore and/or protect social, economic, and cultural values and ecosystem services attributed to coral reef sites, alongside ecological values and functions [18, 19]. Understanding and monitoring this multiplex of values to determine the appropriateness of an ecological intervention and/or to evaluate its success or failure is a significant challenge. As the ecological processes and state of coral reefs fluctuate in response to climatic and other pressures, the interrelated social, economic, and cultural values and processes (i.e., human dimensions) too are dynamic and responsive to myriad pressures. Monitoring public perceptions of proposed or initiated ecological interventions can provide insights to some of those human values and their responsiveness to alterations in the natural environment, as well as the public acceptability of risks associated with the intervention [20–22].

To date, research and monitoring of public perceptions and support for coral reef interventions, including ACR, has largely utilised *ex situ* surveys in which respondents indicate their level of support or concern for specific intervention types (and/or scenarios) that are defined or briefly described [e.g., 20–23] and/or accompanied by imagery [e.g., 17]. Such studies have generally found that most intervention types are broadly supported, with some conditions and predictive factors identified. However, the ways in which such interventions are described or depicted can influence respondent perceptions (e.g., through the use of emotive or normative frames [17]), and studies of this nature can thus be subject to researcher or proponent biases. *In situ* studies can provide an opportunity to better understand people's perceptions and understandings that arise from a first-hand observation and experience, and they can potentially help to 'ground truth' findings derived from *ex situ* studies.

Public, stakeholder and rights-holder (including Traditional Owner) understandings and perceptions of ecological restoration interventions do not arise in an information vacuum. It is therefore incumbent on the proponents of interventions to engage with relevant communities and the wider public to build a shared understanding of the risks and benefits of proposed interventions, as well as to foster opportunities for local communities and enterprises to benefit from such initiatives [24, 25]. Demonstration sites offer one approach to engaging community representatives and the wider public in a meaningful way, enabling sharing of perspectives and knowledge alongside the transparent evaluation of an intervention trial at an early stage and relatively small scale [9, 26].

Our study explores public perceptions of ACR trials *in situ*, involving tourists and local residents visiting sites in the northern Great Barrier Reef (GBR). This work is undertaken as part of a broader suite of research and monitoring activities to (a) better understand public perceptions and support for ACR and other technological interventions to build coral reef resilience, and (b) engage local communities in deliberative risk governance and co-development of benefit pathways from coral reef interventions (within Australia's *Reef Restoration and Adaptation Program*; https://gbrrestoration.org/program/engagement/).

## Case study context: ACR trials at tourism sites in the northern Great Barrier Reef

In the GBR, ACR trials have expanded rapidly since mass coral bleaching events in 2016 and 2017, supported by changes in Marine Park management policy to enable restoration and other interventions [9, 27, 28]. Such field trials are conducted under a Marine Parks permit for scientific research and are led by a range of organisations including scientific institutions, marine tourism businesses, and not-for-profit and non-government organisations [29]. Most ACR trials occupy areas of reef substrate less than 100m$^2$ and many are deployed in proximity to tourism facilities, including moorings and pontoons that are visited regularly by Reef tourism operations [30].

Numerous types of ACR device are deployed in field trials across the Great Barrier Reef (see examples in Fig 1). Our study explored visitor perceptions associated with three device types present at reef tourism sites in the Cairns-Port Douglas region of the GBR Marine Park (Fig 2). These included: (1) *'Reef Stars'* (produced by the *Mars Assisted Reef Restoration System*$^{TM}$ [31])–a modular system of hexagonal metal frames with live coral fragments attached, positioned and interlocked on the substrate (usually coral rubble) to provide a stable platform for corals to grow, (2) *coral nursery frames* (used by the *Coral Nurture Program*; https://www.coralnurtureprogram.org/)—a metal grill suspended above the substrate, used to attach and grow coral fragments to a suitable size prior to out-planting, and (3) *coral seeding devices* (CSD; engineered by the Australian Institute of Marine Science [32])–a modular and mass-producible ceramic housing designed to hold and protect juvenile corals that had spawned and settled in an aquarium, prior to being deployed on reef substrate. Other types of ACR devices were in use in the region at the time of our study, including *Coralclip*(s)® [33], *coral nursery trees* (used by the *Reef Restoration Foundation*; https://rrf.org.au/; also shown in Fig 1), and *rubble-stabilising mesh*. However, these other devices were either not present at our study sites or were deemed unlikely to be observed by enough visitors during our study's timeframe to produce meaningful results.

The presence of ACR devices at reef tourism sites presents opportunities for tourists and other visitors to observe and learn about them. Crew from the tourism vessels typically present a range of educational information to passengers about coral reef ecology and threats, as well as their involvement in local reef stewardship, including ACR activities, with an explanation of

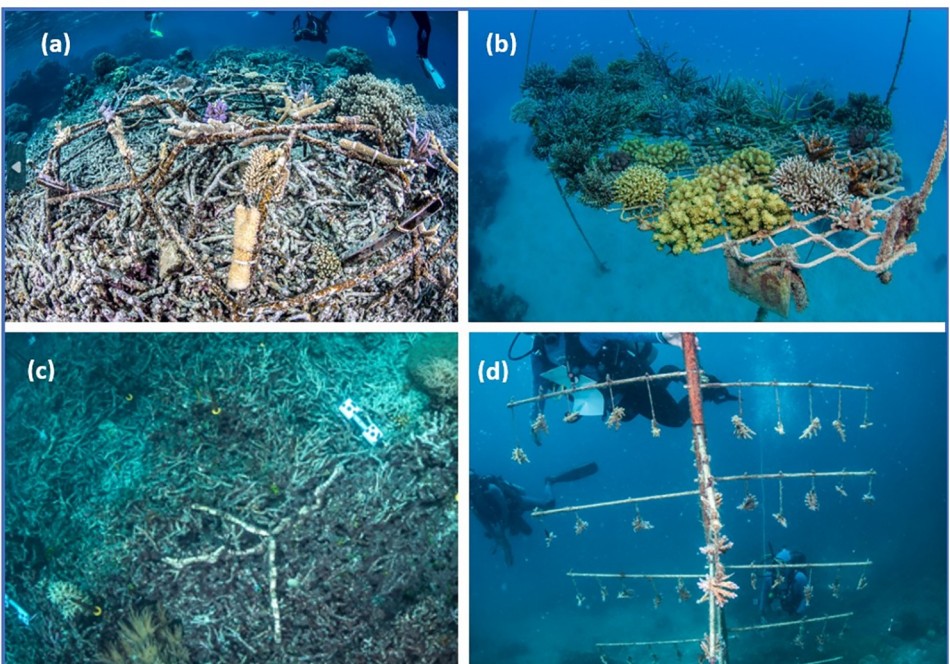

**Fig 1. Underwater photographs of assisted coral recovery (ACR) devices deployed at reef study sites.** Panels a–c show: (a) '*Reef Stars*', with coral fragments attached, deployed over a coral rubble field, (b) a *coral nursery frame*, suspended in the water column to aid rapid re-growth of coral fragments, and (c) *coral seeding devices (CSDs)*, which hold and protect juvenile corals, allowing them to grow on a range of substrates, including coral rubble. Panel (d) shows *coral trees* (not included in the study), used for growing coral fragments that are later re-attached to the reef.

the devices that are observable at their site(s) (authors' pers obs.). It is hypothesised that Reef visitors' observation of ACR devices *in situ* does not negatively affect visitors' satisfaction with their coral reef experience, and if accompanied by appropriate educational information, such experiences may help to raise awareness and support for ACR and other coral reef protection efforts. However, these hypotheses have not, to our knowledge been previously empirically tested.

Multiple factors are known to influence coral reef visitors' satisfaction, including (and not limited to) the weather, the visited site's physical and aesthetic characteristics (e.g. water clarity, coral assemblages, fish abundance), facilities and infrastructure, tour service quality and hospitality, cultural and personal factors (including seasickness), prior experience and expectations, and social interactions [35–39].

## Study aims

Using a visitor survey, our case study sought to assess the influence of seeing ACR devices *in situ* on Reef visitors' experiences and risk perceptions about the use of such devices. Specifically, we investigated whether the observation of ACR devices on coral reef sites had any effect (positive or negative) on Reef visitors': (a) overall satisfaction with their Reef trip, (b) perceptions of the aesthetic beauty of the Reef site(s) visited, (c) concern about the future health of the Reef, and (d) concern about the use of ACR devices on the Reef. We compared the relative effect of *seeing ACR devices* to other potential factors of influence, including the weather conditions, visitor demography (age and local residence/visitor status), their self-assessed level of knowledge and experience of coral reefs, the perceived condition/health of the reef site(s)

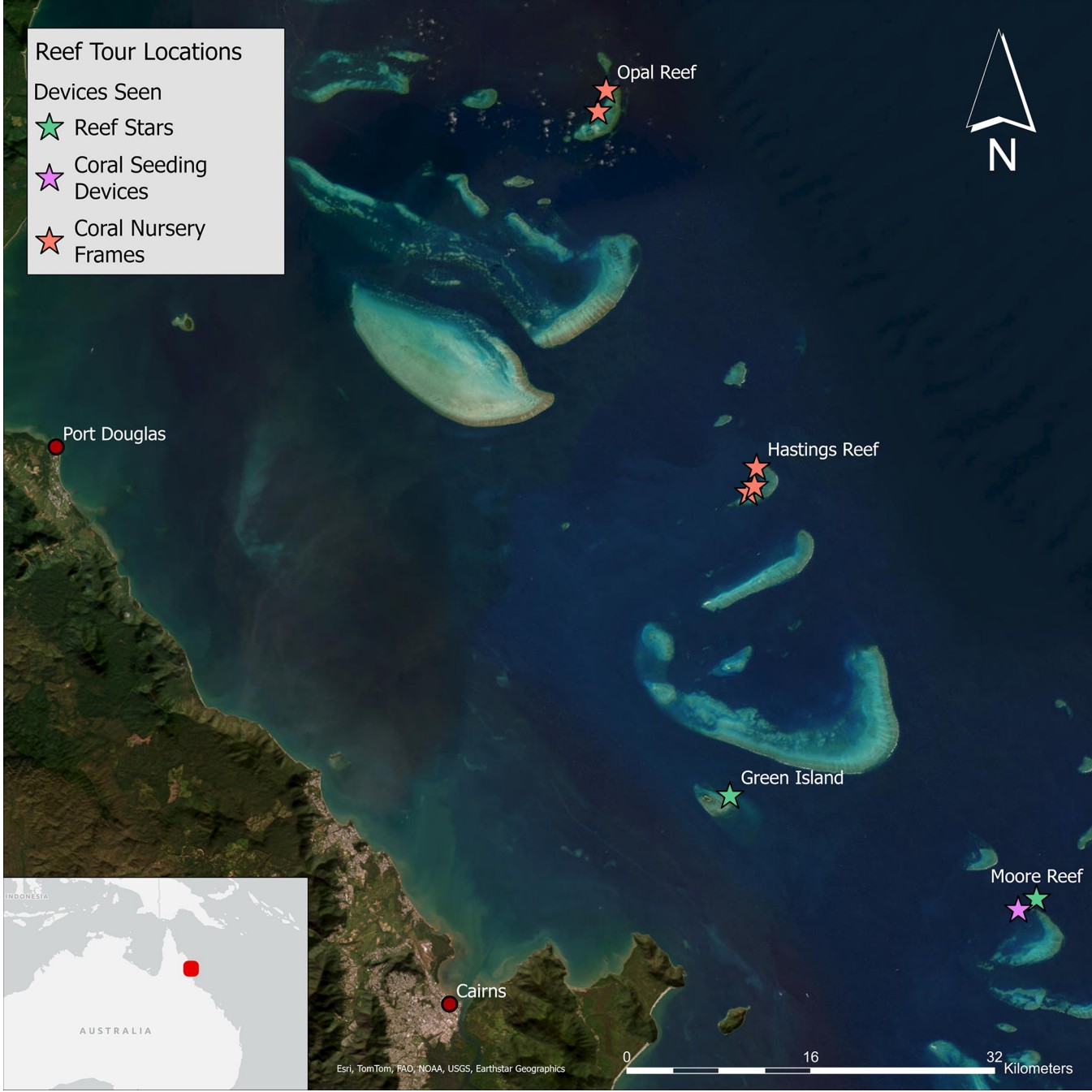

**Fig 2. Study region with approximate locations of reef tourism sites sampled, at which specific types of assisted coral recovery devices were deployed.** Map created using ArcGIS software from ESRI [34]. Basemaps supported by ESRI and reprinted under the ESRI Master License Agreement, CC BY 4.0. Original copyright ESRI 2024.

visited, and the perceived quality of educational information that accompanied the visitors' Reef experiences.

Findings from this case study are relevant to ACR practitioners and proponents, including scientists, Traditional Owners, coral reef managers and marine tourism operators who are concerned about public visibility of ACR trials, Reef visitor perceptions and support for ACR,

and the potential value of demonstration sites that seek to enhance public understanding and support for resilience-building coral reef interventions.

## Materials and methods

### Survey design and reef sites

As part of a collaborative partnership approach, our visitor survey, based on a self-administered questionnaire, was co-designed with input from several Reef tourism operators in the Cairns and Port Douglas region who were involved in ACR trials at their reef sites. To enable robust statistical comparisons of the potential effect of observing ACR devices *in situ*, we sought to collect equivalent samples of visitors who saw ACR devices on their Reef day trip and visitors who did not see ACR devices.

Surveys were conducted on five vessels that visited one or more reef sites (8 sites in total) at which ACR devices were deployed. Due to the varying spatial extent of each reef site (up to several hectares), varied topography, relative space used by the different ACR trials (up to approximately 500m$^2$ at one reef site) and their varying proximity to the vessel or other tourist facilities, not all passengers were able to observe ACR devices on their trip, regardless of the vessel. At some reef sites, the ACR devices were only visible to a small proportion of passengers, for example, those who took part in a guided snorkel or semi-submersible tour. Thus, on each trip, a varying proportion of passengers observed ACR devices *in situ*.

Images and a description of each type of ACR device were provided on the cover page of each questionnaire for respondents' reference (as per Fig 1). *Coral nursery frames* were deployed at five different reef sites (two sites at Opal Reef and three sites at Hastings Reef), while *Reef Stars* were deployed at two sites (one in the vicinity of Green Island, and one at Moore Reef) and *CSDs* were deployed at one reef site only (Moore Reef; see Fig 2). We note that at the time of the study, CSDs could only be viewed by small groups of visitors who took part in an 'adventure snorkel safari' from one of the participating vessels. These snorkellers were typically accompanied by a tour guide who led them to the location of the CSDs and then pointed them out as a feature of interest. Even then, the devices were occasionally difficult to see amidst the coral rubble, due to their depth (approx. 6m) and their accumulation of algae over time. Other devices were typically observed by reef visitors who participated in snorkelling, scuba diving, and/or guided semi-submersible/glass-bottom boat tours.

### Research participants, sampling protocol and ethics

Our target population was English speaking GBR visitors (noting our language capability limitations) over 18 years of age, who took part in a GBR day trip on one of the five selected vessels, visiting site(s) in the Cairns-Port Douglas region at which ACR devices were deployed, during September and October 2023. The survey was timed over the school holidays to maximise the number of potential participants over the sampling timeframe. Questionnaires were distributed in paper form by a member of the research team and/or a member of the vessel crew on the return journey from the Reef site(s) to Cairns or Port Douglas. They were typically distributed to passengers in the main saloon, inside the vessel, which provided seating and tables amenable to completing a questionnaire. Researchers and/or crew would make a brief announcement to passengers about the purpose and terms of the survey before inviting passengers to participate and distributing questionnaires. Completion of the questionnaire took between five and ten minutes for most respondents.

Prior to commencement, the study was reviewed and approved by CSIRO's Social Science Human Research Ethics Committee (Approval #131/21) in accordance with Australia's National Statement on Ethical Conduct in Human Research (2007). Consent to participate

was given verbally. Respondents were asked if they would like to participate, and if they agreed, they were provided a paper copy of the survey and a pen. Details about the purpose of the study, the handling and uses of the data, respondents' anonymity and other ethical considerations were provided on the survey cover sheet, and all respondents' participation was entirely voluntary. Additionally, the survey cover sheet stated that respondents were free to withdraw by stopping the survey at any time. Consent to participate was implied if respondents completed and returned their questionnaire to the researcher on board. Approximately 50 incomplete, returned questionnaires were excluded from the study.

## Survey questions

Survey questions focussed on Reef visitors' experience during their day trip to one or more GBR sites. Likert-style ten-point rating scales were provided for responses to questions about: visitors' *overall trip satisfaction* (1 = 'very dissatisfied'; 10 = 'extremely satisfied'), *perceptions of the visual beauty* of the reef site(s) they visited (1 = 'very ugly, unpleasant'; 10 = 'exceptionally beautiful'), *perceptions of the overall condition* of the reef site(s) they visited (1 = 'very poor, unhealthy'; 10 = 'excellent condition'), *concern about the use of ACR devices* on the Reef and *concern about the future health of the Reef* (1 = 'not concerned at all'; 10 = 'extremely concerned'), and their evaluation of the quality of information they received about ACR on their trip (1 = 'very poor quality'; 10 = 'excellent quality').

Additional questions solicited information about respondents' demography (country of residence, age in categories, gender), their self-assessed *level of knowledge and experience of coral reefs* (10-point rating scale; 1 = 'novice, low familiarity with coral reefs, health and ecology'; 10 = 'expert, highly experienced and knowledgeable about coral reef health and ecology'), their *perception of the weather* on the day of their trip (5 categories; 1 = 'terrible', 2 = 'poor', 3 = 'fair', 4 = 'good', 5 = 'amazing'), their activities whilst visiting the Reef on the day, their prior awareness of ACR on the GBR, and whether they had seen any ACR devices on their trip to the GBR that day, including the four types of devices, as explained on the cover sheet.

We used ten-point Likert-type rating scales for appropriate questions, as methodological experiments have shown they offer higher statistical discriminating power and test-retest reliability than shorter scale lengths, are less prone to response-style biases than five and seven-point scales, and are preferred by most survey respondents over other scale lengths [40, 41]. We also used categorical responses (with fewer options) for some questions to reduce respondents' cognitive burden (e.g., when estimating the weather conditions).

## Data analyses

We used R Statistical Software (v4.3.1) [42] to perform a series of ordinal logistic regression analyses with respondents' numerical ratings given in response to the questions outlined above. Assumptions relevant to the ordinal logistic regression tests performed in this study include proportional odds, goodness of fit, and no multicollinearity. We investigated the latter by calculating variance inflation factors (VIF), where values less than five were deemed acceptable. We used a Brant test to check for the assumption of proportional odds (also known as parallel lines) and the Lipsitz test to assess goodness of fit [43, 44]. We also ran a rank-based, non-parametric Kruskal-Wallis H test to examine potential differences in the distributions of ratings of *concern about the use of ACR devices* between groups of respondents who had seen different types of device.

## Sample description

The survey resulted in a total of 708 eligible respondents from five participating vessels, 346 of whom (49%) reported that they saw one or more ACR devices on the reef sites they visited that

day. Of those who saw ACR devices, 165 (23%) reported seeing *Reef Stars*, while 164 (23%) reported seeing *coral nursery frames*, and 56 (8%) reported seeing *CSDs*. From one of the vessels (Vessel A) it was possible to observe both *Reef Stars* and *CSDs* on the same reef trip, and 39 respondents from this vessel reported seeing both. The number of respondents from each of the five vessels (and proportion that saw any ACR device in %) was: Vessel A: n = 262 (64% saw ACR device(s)), Vessel B: n = 151 (42%), Vessel C: n = 150 (9%), Vessel D: n = 91 (85%), and Vessel E: n = 54 (44%). We note that 20 respondents (6 from Vessel B, 7 from Vessel C, and 7 from Vessel D) reported that they had seen *CSDs* in addition to other types of device, despite *CSDs* only being present at the site visited by Vessel A.

From the total sample, 413 respondents (58%) identified as female and 282 (40%) identified as male. Alternate gender response options included 'other' (n = 3 respondents) and 'prefer not to say' (n = 2). A further eight respondents left the field blank. Due to their low occurrence, the latter two response options were omitted from our regression tests of gender as a predictor variable. The category representing the median age of respondents was 35 to 44 years. Respondents came from 33 different countries; however, 423 (60%) were Australian and 55 (8%) were residents of the Great Barrier Reef catchment region, living between Bundaberg and Cape York, east of the Great Dividing Range (indicated via postcode). Other countries with the largest proportion of respondents included the United Kingdom (n = 56; 8% of the total sample), the USA (n = 54; 8%), China (n = 31; 4%), Germany (n = 24; 3%), and New Zealand (n = 21; 3%).

## Results

### Satisfaction with reef trip

The mean rating of *overall trip satisfaction* for the total sample was 8.67 out of ten (±0.048 SE; n = 708), with 60% of respondents (n = 422) providing a rating of nine or ten, indicating they were 'extremely satisfied'. Ordinal regression tests investigated potential relationships between respondents' ratings of their *overall trip satisfaction* and *seeing ACR devices*, alongside other factors including respondents': (i) *age* category, (ii) *place of residence* (i.e., comparing residents of the GBR catchment region with visitors to the region), (iii) self-rated level of *knowledge and experience of coral reefs*, (iv) perceived *quality of information* they received about ACR on their trip, (v) perceived *condition of the reef site(s)* they visited, and (vi) perceived *weather conditions* during their Reef day trip (Fig 3A).

Significant positive relationships with *overall trip satisfaction* were found for (i) *weather conditions* (which had the strongest effect size; β = 0.777; z = 6.366; p = 0.000), (ii) perceived *condition of the reef site(s)* (β = 0.555; z = 6.892; p = 0.000), and (iii) perceived *quality of information* received on the trip (β = 0.193; z = 3.345; p = 0.001). There was no significant relationship between *overall trip satisfaction* and *seeing ACR devices* (β = 0.151; z = 0.706; p = 0.480), nor any relationship with respondents' *residency in the GBR region* (β = 0.128; z = 0.418; p = 0.676), *age* (β = 0.021; z = 0.258; p = 0.797), *knowledge and experience of coral reefs* (β = -0.013; z = -0.257; p = 0.797), or *gender* (β = -0.280; z = -1.308; p = 0.191; Fig 3A).

### Perceived beauty of reef sites

The mean rating for *visual beauty of the reef site(s)* for the total sample was 8.21 out of ten (±0.059 SE; n = 708), with 46% of respondents (n = 329) providing a rating of nine or ten, indicating they perceived the reef site(s) to be 'exceptionally beautiful'. Regression test results revealed no significant relationship between respondents' rating of the *visual beauty of the reef site(s)* they visited and *seeing ACR devices* (β = 0.188; z = 0.867; p = 0.386), nor with their *residency in the GBR region* (β = 0.341; z = 1.046; p = 0.295), *gender* (β = 0.250; z = 1.159; p = 0.246), or perceived *weather conditions* (β = 0.184; z = 1.568; p = 0.117; Fig 3B). However,

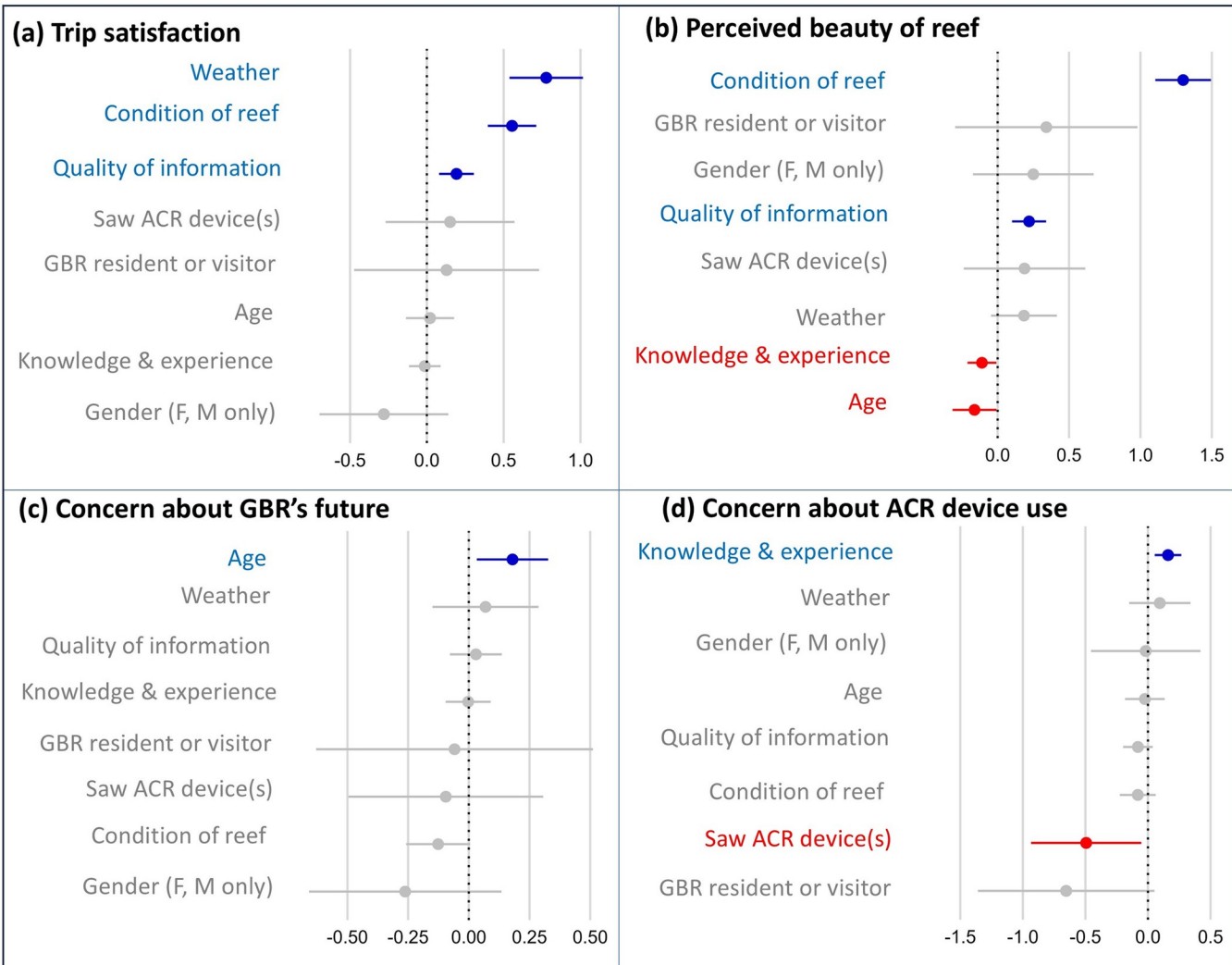

**Fig 3.** Ordinal regression test results plotting survey respondents' ratings of (a) *overall trip satisfaction*, (b) *perceived beauty of the reef site(s)* they visited, *(c) concern about the future health of the Great Barrier Reef*, and (d) *concern about the use of assisted coral recovery (ACR) devices* on the reef, with potential predictor variables, including *seeing ACR devices* at the reef site(s) they visited. Panels a–d show regression coefficients [dots] and standard error [SE] bars, indicating statistical significance of the relationship where the SE bar does not intersect with zero [dotted line] (n = 708). Blue colouring indicates a significant positive relationship and red colouring indicates a significant negative relationship.

significant positive relationships were found with respondents' perceived *condition of the reef site(s)* (with the strongest effect size; β = 1.299; z = 13.084; p = 0.000), and respondents' perceived *quality of information* received on the trip (β = 0.220; z = 3.625; p = 0.000), while significant negative relationships were found with respondents' *knowledge and experience of coral reefs* (β = -0.110; z = -2.139; p = 0.032) and their *age* (β = -1.62; z = -2.074; p = 0.038)–i.e., older respondents and those with greater *knowledge and experience of coral reefs* were slightly more likely to give lower ratings for the *visual beauty of the reef site(s)*.

## Concern about future health of the GBR

The mean rating for *concern about the future health of the GBR* for the total sample was 8.06 out of ten (±0.076 SE; n = 695), and 48% of respondents (n = 331) gave a rating of nine or ten,

indicating they were 'extremely concerned' about the future health of the GBR. Only one variable, respondent *age*, was found to have a significant positive relationship with *concern about the future health of the GBR* (β = 0.180; z = 2.404; p = 0.016)–i.e., older respondents were more likely to give higher ratings of concern (Fig 3C). No significant relationships were found between respondents' ratings for *concern about the future health of the GBR* and other variables tested, including perceived *weather conditions* (β = 0.068; z = 0.619; p = 0.536), perceived *quality of information* received on the trip (β = 0.029; z = 0.537; p = 0.591), respondents' *knowledge and experience of coral reefs* (β = -0.002; z = -0.053; p = 0.958), *residency in the GBR region* (β = -0.059; z = -0.202; p = 0.840), *seeing ACR devices* (β = -0.095; z = -0.463; p = 0.643), perceived *condition of the reef site(s)* visited (β = -0.126; z = -1.882; p = 0.060), or respondent *gender* (β = -0.262; z = -1.295; p = 0.195; Fig 3C).

## Concern about the use of ACR devices

The mean rating for *concern about the use of ACR devices on the Reef* for the total sample was 2.69 out of ten (±0.095 SE; n = 684). Five per cent of respondents (n = 32) gave a rating of nine or ten, indicating they were 'extremely concerned' about the use of ACR devices, while 65% (n = 446) gave a rating of one or two, indicating they were 'not concerned at all'. Ordinal regression tests revealed a significant negative relationship between respondents' *concern about the use of ACR devices* on the reef and *seeing ACR devices*–i.e., respondents who saw ACR devices deployed at the reef sites they visited were more likely to express lower levels of concern about their use (β = -0.494; z = -2.198; p = 0.028). There was also a small but significant positive relationship between respondents' ratings of *concern about the use of ACR devices* and their self-rated *level of knowledge and experience of coral reefs* (β = 0.160; z = 2.967; p = 0.003)–i.e., respondents with greater *knowledge and experience of coral reefs* were slightly more likely to give higher ratings of concern (Fig 3D).

Other variables showed no significant relationship with concern about the use of ACR devices, including the *weather conditions* (β = 0.095; z = 0.760; p = 0.447), respondent *gender* (β = -0.018; z = -0.080; p = 0.936), *age* (β = -0.024; z = -0.302; p = 0.762), perceived *quality of information* (β = -0.080; z = -1.326; p = 0.185), perceived *condition of the reef site(s)* (β = -0.081; z = -1.112; p = 0.266) and respondents' *residency in the GBR region* (β = -0.654; z = -1.815; p = 0.070; Fig 3D).

## Differences between device types

We compared mean ratings of *concern about the use of ACR devices* between groups of respondents who saw different device types, including *CSDs* (Group A; n = 56 including the 39 who had seen both CSDs and Reef Stars; x̄ = 2.24 ± 0.272 SE), *Reef Stars* (Group B; n = 126 who had only seen Reef Stars; x̄ = 2.55 ± 0.199 SE), and *coral nursery frames* (Group C; n = 164; x̄ = 2.19 ± 0.185 SE). An Independent Samples Kruskal-Wallis one-way ANOVA found no significant difference between the three groups' distributions of ratings of *concern about the use of ACR devices* ($X^2$ = 5.63, p = 0.06, df = 2).

## General perceptions of ACR devices

Respondents were asked: "*Do you think this kind of assisted coral recovery on the Great Barrier Reef is worthwhile?*" Short response categories (tick-boxes) included 'yes,' 'no' and 'unsure', and a space was provided for respondents to write a brief explanation for their answer. Of the 699 respondents who answered the question, 86% (n = 599) indicated 'yes', while 14% (n = 95) indicated 'unsure' and less than 1% (n = 5) indicated 'no' in the short response. A total of 291 respondents provided a short text explanation for their response. Only two of the five

respondents who indicated 'no' to the preceding question provided a brief statement. Those statements were: "*Seem completely innocuous*", and "*A really good idea*"–the latter statement being incongruous with their 'no' response.

Text responses among respondents who indicated they were unsure about whether this kind of ACR on the GBR is worthwhile (n = 39 in total) were more varied. Example statements included: "*I don't know enough to judge*", "*It doesn't hurt the experience*", "*If they help the reef then I am not concerned*", "*Could this disturb the natural balance of the reef?*", "*They are a sign of the times*", "*If they are meant to help reef recovery, then yes I am concerned about how effective they are*", "*The ones we saw were anchored in sand and there was no problem that we could foresee*", and "*These devices, while useful, will not offset climate change*".

Statements provided by respondents who thought this kind of ACR was worthwhile (n = 250 in total) were similarly varied. Example statements included: "*Not sure if there's any drawbacks of using these devices*", "*Yes, these devices are important for the regrowth*", "*I have seen a documentary on coral reef repopulation*", "*Whatever helps the reef is a good thing*", "*We were concerned about leaving nails/wire in the water*", "*Seems benefits would outweigh any risks*", "*Low impact in appearance*", "*Small amount of concern as they are introduced man-made products. But they break down over time which is good*", and "*I believe you have to do as much as possible to recover the reef*".

## Discussion

Key findings from our study included: (1) no significant influence of seeing ACR devices at reef sites on reef visitors' overall trip satisfaction, perceptions of the reef's visual aesthetic beauty, and concern about the future health of the GBR, and (2) concerns about the use of ACR devices were significantly lower among visitors who observed them at reef site(s).

Our findings have implications for those involved in planning, designing, and delivering ACR initiatives and other coral reef protection interventions. Public concerns about the deployment of human-made devices in a natural coral reef setting, particularly within a World Heritage Area inscribed for its natural values, are likely to be an important consideration that influences decisions about the location of such deployments and the potential value of demonstration sites. While most survey respondents indicated a low level of concern overall about the use of ACR devices on reef sites, we note that small groups of opponents can wield significant social and political influence that can undermine political support for environmental interventions and developments [45]. As new techniques and types of intervention are developed and tested, and as the deployment of intervention technologies increases in spatial scale, proponents and decision makers may wish to foster greater public understanding of either positive or negative impacts of such interventions to Reef values by providing greater opportunities for public observation to build familiarity.

Reef tourism operators are already playing a valuable role in influencing public perceptions of ACR, by enabling Reef visitors to observe the devices *in situ*, and through their delivery of accompanying educational information. While an evaluation of the content and delivery of such information was beyond our study's scope, our results indicate that the perceived quality of interpretive information has a significant influence on visitor satisfaction and aesthetic perceptions of the reef site(s) they visited (Fig 3A and 3B). These insights are valuable for shaping the design and delivery of the rapidly evolving field of ACR in a way that is more alert to social, economic, and cultural values alongside ecological values. Further research that involves co-designing and evaluating demonstration sites for ACR, including the content and delivery of interpretive material that accompanies reef visitors' exposure to ACR initiatives, would seem worthwhile. In addition to raising public awareness of Reef threats, ACR and other Reef

protection interventions, the provision of high-quality interpretive material may also enhance Reef visitors' satisfaction, benefitting tourism businesses. Better understanding of these benefits to tourism operators could also help inform positive progress on the types of service provision or collaborative business models needed to help scale restoration activities.

### Limitations, knowledge gaps and further research

The influence of other factors such as weather, service quality, and reef condition (among other things) on reef visitors' trip satisfaction are well documented [e.g., 36]. However, our study is the first to compare the relative effect of observing ACR devices *in situ* with such factors. While our analysis of survey data finds no evidence that the presence of ACR devices detracts from Reef visitor experiences, we note that the ACR deployments in this study were confined to relatively small areas (and in some cases were hard to find, due to algal growth or coral cover), among an otherwise natural and unaltered coral reef setting. Larger scale deployments to restore damaged or degraded reefs may elicit a different visitor and/or public reaction, particularly if the setting is dominated by artificial assemblages.

Our study was limited in scope, recruiting only English-speaking respondents, and using a limited number of survey questions to enable short completion times and minimise its potential intrusiveness for Reef visitors. The GBR receives a diversity of international visitors each year, and future studies that include non-English speaking participants may identify other demographic and cultural factors that influence reef visitor perceptions of ACR devices. Similarly, our study was limited to only eight Reef sites in a relatively small region (Fig 2), and it was beyond our ability to quantify or control the environmental conditions and myriad other *in situ* variables that may have contributed to respondents' experiences and survey responses. Further research and monitoring of public, stakeholder, and rights-holder perceptions and support are therefore needed to accompany different types of interventions at varying scales, and longitudinal studies could enable the assessment of any enduring impacts of ACR on public understanding and acceptance, as well as on coral reef health. Demonstration sites can help to facilitate such studies while providing opportunities for meaningful engagement between ACR proponents and communities. These types of integrated, place-based, and accessible monitoring strategies that include tracking site visitors' and others' perceptions of ACR devices in addition to ecological outcomes, we believe, will be increasingly important for restoration proponents in improving transparency and accountability and therefore public, regulator and investor support.

### Conclusion

Overall, respondents expressed a very high to extreme level of concern about the future health of the GBR, and most considered ACR efforts to be worthwhile. We note that in the months following our survey's completion the GBR was subjected to a fifth widespread coral bleaching event since 2016, as part of a global coral bleaching event attributed to unprecedented ocean temperatures [4, 46]. Considering the GBR's importance as an iconic ecosystem for Australians and the international community [47, 48], its dire outlook in the face of a rapidly changing climate [49, 50], and public sentiment responses to recent major coral bleaching events [51, 52], there is a strong impetus for the rapid advancement and scaling of ACR and other coral reef interventions. However, while public support for specific coral reef interventions is not guaranteed, there is, at the same time, a desire amongst publics to retain a sense of individual and collective efficacy for action, and a sense of hope for the future [53, 54]. Studies such as this provide a clear signal that both these ambitions remain possible.

## Acknowledgments

We acknowledge the Traditional Owners and custodians of the sea country on which our study took place, including the Gunggandji, Yirrganydji, and Eastern Kuku Yalanji peoples.

This study was conducted as part of research activities by the Reef Restoration and Adaptation Program's (RRAP) *Best Practice Stakeholder and Traditional Owner Engagement Subprogram* (https://gbrrestoration.org/program/engagement/). The study would not have been possible without in-kind support from partners in the Cairns and Port Douglas region, including *GBR Biology* (Experience Co Ltd), the *Reef Restoration Foundation*, *Passions of Paradise*, *Wavelength Reef Cruises*, *Down Under Cruise and Dive*, *Quicksilver Connections*, TropWATER, and the Cairns–Port Douglas Reef Hub (https://www.reefhub.com.au/).

The scientific results and conclusions, as well as any views or expressed herein, are those of the authors and do not necessarily reflect the views of the Australian Government or the Minister for the Environment. The authors declare no conflict of interest.

## Author Contributions

**Conceptualization:** Matthew I. Curnock, Katherine Chartrand, Eric E. Fisher, Rebecca Forster, Stewart Lockie, Jennifer Loder, Abigail Scott, Bruce Taylor.

**Data curation:** Matthew I. Curnock, Rhea Arya, Emilee Chamberland.

**Formal analysis:** Matthew I. Curnock, Emilee Chamberland, Danielle Nembhard.

**Funding acquisition:** Stewart Lockie, Bruce Taylor.

**Investigation:** Matthew I. Curnock, Rhea Arya, John Edmondson, Eric E. Fisher, Abigail Scott, Jasmina Uusitalo.

**Methodology:** Matthew I. Curnock, Rhea Arya, Katherine Chartrand, John Edmondson, Eric E. Fisher, Abigail Scott, Jasmina Uusitalo.

**Project administration:** Matthew I. Curnock.

**Supervision:** Matthew I. Curnock.

**Validation:** Matthew I. Curnock, Emilee Chamberland.

**Visualization:** Matthew I. Curnock, Emilee Chamberland, Danielle Nembhard.

**Writing – original draft:** Matthew I. Curnock, John Edmondson, Eric E. Fisher, Jennifer Loder, Danielle Nembhard.

**Writing – review & editing:** Matthew I. Curnock, Rhea Arya, Emilee Chamberland, Katherine Chartrand, John Edmondson, Eric E. Fisher, Rebecca Forster, Stewart Lockie, Jennifer Loder, Danielle Nembhard, Abigail Scott, Bruce Taylor, Jasmina Uusitalo.

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
