## [Decision Letter · Decision Letter 0]

9 Sep 2024

PONE-D-24-22940Reef visitors’ observation of assisted coral recovery devices in situ reduces concern about their usePLOS ONE

Dear Dr. Curnock,

Thank you for submitting your manuscript to PLOS ONE. After careful consideration, we feel that it has merit but does not fully meet PLOS ONE’s publication criteria as it currently stands. Therefore, we invite you to submit a revised version of the manuscript that addresses the points raised during the review process.

We look forward to receiving your revised manuscript.

Kind regards,

Nadeem Nazurally, Ph.D

Academic Editor

PLOS ONE

Journal Requirements: When submitting your revision, we need you to address these additional requirements. 1. Please ensure that your manuscript meets PLOS ONE's style requirements, including those for file naming. The PLOS ONE style templates can be found at https://journals.plos.org/plosone/s/file?id=wjVg/PLOSOne_formatting_sample_main_body.pdf and https://journals.plos.org/plosone/s/file?id=ba62/PLOSOne_formatting_sample_title_authors_affiliations.pdf 2. Please note that PLOS ONE has specific guidelines on code sharing for submissions in which author-generated code underpins the findings in the manuscript. In these cases, all author-generated code must be made available without restrictions upon publication of the work. Please review our guidelines at https://journals.plos.org/plosone/s/materials-and-software-sharing#loc-sharing-code and ensure that your code is shared in a way that follows best practice and facilitates reproducibility and reuse. 3. In the ethics statement in the Methods, you have specified that verbal consent was obtained. Please provide additional details regarding how this consent was documented and witnessed, and state whether this was approved by the IRB. 4. Thank you for stating the following financial disclosure: "This study was conducted as part of research activities by the Reef Restoration and Adaptation Program’s (RRAP) Best Practice Stakeholder and Traditional Owner Engagement Subprogram (https://gbrrestoration.org/program/engagement/), with funding provided by the partnership between the Australian Government’s Reef Trust and the Great Barrier Reef Foundation, delivered in partnership with CSIRO, the Australian Institute of Marine Science, James Cook University, and the Queensland University of Technology. "  Please state what role the funders took in the study.  If the funders had no role, please state: ""The funders had no role in study design, data collection and analysis, decision to publish, or preparation of the manuscript."" If this statement is not correct you must amend it as needed. Please include this amended Role of Funder statement in your cover letter; we will change the online submission form on your behalf. 5. Please expand the acronym “RRAP ENG” (as indicated in your financial disclosure) so that it states the name of your funders in full. This information should be included in your cover letter; we will change the online submission form on your behalf. 6. We note that Figure 2 in your submission contain [map/satellite] images which may be copyrighted. All PLOS content is published under the Creative Commons Attribution License (CC BY 4.0), which means that the manuscript, images, and Supporting Information files will be freely available online, and any third party is permitted to access, download, copy, distribute, and use these materials in any way, even commercially, with proper attribution. For these reasons, we cannot publish previously copyrighted maps or satellite images created using proprietary data, such as Google software (Google Maps, Street View, and Earth). For more information, see our copyright guidelines: http://journals.plos.org/plosone/s/licenses-and-copyright. We require you to either (1) present written permission from the copyright holder to publish these figures specifically under the CC BY 4.0 license, or (2) remove the figures from your submission: A. You may seek permission from the original copyright holder of Figure 2 to publish the content specifically under the CC BY 4.0 license.   We recommend that you contact the original copyright holder with the Content Permission Form (http://journals.plos.org/plosone/s/file?id=7c09/content-permission-form.pdf) and the following text:“I request permission for the open-access journal PLOS ONE to publish XXX under the Creative Commons Attribution License (CCAL) CC BY 4.0 (http://creativecommons.org/licenses/by/4.0/). Please be aware that this license allows unrestricted use and distribution, even commercially, by third parties. Please reply and provide explicit written permission to publish XXX under a CC BY license and complete the attached form.” Please upload the completed Content Permission Form or other proof of granted permissions as an ""Other"" file with your submission. In the figure caption of the copyrighted figure, please include the following text: “Reprinted from [ref] under a CC BY license, with permission from [name of publisher], original copyright [original copyright year].” B. If you are unable to obtain permission from the original copyright holder to publish these figures under the CC BY 4.0 license or if the copyright holder’s requirements are incompatible with the CC BY 4.0 license, please either i) remove the figure or ii) supply a replacement figure that complies with the CC BY 4.0 license. Please check copyright information on all replacement figures and update the figure caption with source information. If applicable, please specify in the figure caption text when a figure is similar but not identical to the original image and is therefore for illustrative purposes only.The following resources for replacing copyrighted map figures may be helpful: USGS National Map Viewer (public domain): http://viewer.nationalmap.gov/viewer/The Gateway to Astronaut Photography of Earth (public domain): http://eol.jsc.nasa.gov/sseop/clickmap/Maps at the CIA (public domain): https://www.cia.gov/library/publications/the-world-factbook/index.html and https://www.cia.gov/library/publications/cia-maps-publications/index.htmlNASA Earth Observatory (public domain): http://earthobservatory.nasa.gov/Landsat: http://landsat.visibleearth.nasa.gov/USGS EROS (Earth Resources Observatory and Science (EROS) Center) (public domain): http://eros.usgs.gov/#Natural Earth (public domain): http://www.naturalearthdata.com/ 7. Please review your reference list to ensure that it is complete and correct. If you have cited papers that have been retracted, please include the rationale for doing so in the manuscript text, or remove these references and replace them with relevant current references. Any changes to the reference list should be mentioned in the rebuttal letter that accompanies your revised manuscript. If you need to cite a retracted article, indicate the article’s retracted status in the References list and also include a citation and full reference for the retraction notice.

**Additional Editor Comments:**

Kindly go through the Reviewers comments and address same.

Reviewers' comments:

Reviewer's Responses to Questions

**Comments to the Author**

1. Is the manuscript technically sound, and do the data support the conclusions?

Reviewer #1: Yes

Reviewer #2: Yes

2. Has the statistical analysis been performed appropriately and rigorously? 

Reviewer #1: Yes

Reviewer #2: Yes

3. Have the authors made all data underlying the findings in their manuscript fully available?

Reviewer #1: Yes

Reviewer #2: Yes

4. Is the manuscript presented in an intelligible fashion and written in standard English?

Reviewer #1: Yes

Reviewer #2: Yes

5. Review Comments to the Author

Reviewer #1: Review of the manuscript titled: “Reef visitors’ observation of assisted coral recovery devices in situ reduces concern about their use”

General understanding of the manuscript

The manuscript presents a study on the impact of Assisted Coral Recovery (ACR) devices on visitor experiences and perceptions at the Great Barrier Reef (GBR). The study evaluates the influence of observing ACR devices on overall trip satisfaction, perceptions of reef beauty, concern about the GBR's future health, and concern about ACR devices themselves.

The manuscript claims (i) no significant influence of ACR devices on overall trip satisfaction, visual beauty perceptions, or concern about the GBR’s health, and a reduced concern about ACR devices among visitors who observed them.

These claims are significant for coral reef management and restoration strategies. Understanding how ACR devices affect public perception and experience is crucial for designing effective reef interventions and fostering public support.

Introduction: The authors place their study within the context of existing literature on visitor perceptions of reef interventions and ACR devices. They appropriately reference studies on factors affecting visitor satisfaction and concern about reef health. The literature review is comprehensive but could benefit from a more detailed discussion of specific studies on public perception of similar reef interventions. Highlighting gaps in existing literature and how this study addresses them would strengthen the context.

Materials and methods: The data are extensive, with large sample sizes and detailed statistical analyses. However, some areas could be improved:

- Supporting Claims: The claim that ACR devices do not significantly impact satisfaction or beauty perceptions is well-supported. However, the authors should consider discussing the potential for different visitor demographics or reef conditions to affect these outcomes.

- Additional Evidence: Including more nuanced analyses, such as interaction effects between different variables, could provide a deeper understanding of the data.

Provide, if possible, additional context or examples where the presence of ACR devices might influence perceptions differently, particularly in larger or more altered reef settings.

About the Protocols and Methods: the methods are appropriate, but detailed protocols for specific analyses or observational criteria are not provided. I recommend the authors to consider including a detailed protocol for the observational aspects of ACR device interactions and visitor perceptions. This would aid in replicability and transparency.

The manuscript is well-organized and written clearly. The structure facilitates understanding for both specialists and non-specialists. But, the clarity of the figure could be improved to ensure that the visual data representations are easily interpretable and well-labeled, especially by the non-specialists, for example, by adding some simple results or other types of illustrations.

The manuscript presents valuable insights into the impact of ACR devices on visitor perceptions at the GBR. However, considering the low concern about ACR devices reported by a significant portion of respondents, it may be beneficial for the authors to address and discuss potential biases or limitations in the survey sample or methodology. Furthermore, addressing the broader implications of the findings in relation to larger-scale interventions and different reef settings would enhance the manuscript.

I recommend encouraging the authors to address the comments provided and resubmit a revised version of the manuscript. The study shows significant potential and could contribute valuable knowledge to the field of coral reef management and restoration.

Reviewer #2: The manuscript clearly highlights that the use of assisted coral recovery (ACR) is minimally disruptive to visitors’ experiences in the Great Barrier Reef (GBR), and may even contribute to increased public acceptance and support for reef restoration projects. Given the rapid expansion of ACR initiatives due to the urgency to address coral reef degradation, the authors demonstrate the existing opportunity to balance ecological interventions and still maintain the quality of visitor experiences. The findings are well supported by the empirical approach employed, comprehensive range of perspectives drawn from an adequate sample size, and regression analysis which adds rigor to the findings.

The authors' hypothesis focused on visitors' concerns regarding Assisted Coral Recovery (ACR) devices and was framed around potential negative reactions or apprehensions from visitors when they observed these devices. The study apparently overlooked the potential positive responses, such as appreciation, curiosity, or even admiration and support for conservation efforts. Exploring both sides would have provided a more balanced view of how ACR devices are perceived and could inform more nuanced conclusions about visitor experiences and their support for coral conservation efforts. In addition, this manuscript would have benefited from a global representation on public perceptions if the authors moved beyond selecting English-Speaking participants only, possible by providing translations, given the diversity of international visitors in the GBR.

Other than that, there were the following minor observations:

1. Page 12, para 1: The decision to use a 10-point Likert scale should be justified, as it seems too coarse particularly over a 5-point or 7-point scale.

2. Page 17, para 1: The analysis shows that respondents who were residents to the GBR were significantly more likely to express a lower level of concern on ACR. However, younger age category showed more concern on ACR observations. The implication of these findings should be emphasized in the discussion, especially on the potential to raise local opportunities for the promotion of ACR interventions.

6. PLOS authors have the option to publish the peer review history of their article (what does this mean?). If published, this will include your full peer review and any attached files.

Reviewer #1: No

Reviewer #2: No

---

## [Author Response · Author response to Decision Letter 0]

16 Oct 2024

In the attached Response to Reviewers document, we have comprehensively addressed all the specific reviewer and editor comments. We thank the reviewers and editor for their helpful feedback.

---

## [Editor Report · Decision Letter 1]

23 Oct 2024

Reef visitors’ observation of assisted coral recovery devices in situ reduces concern about their use

PONE-D-24-22940R1

Dear Dr. Curnock,

We’re pleased to inform you that your manuscript has been judged scientifically suitable for publication and will be formally accepted for publication once it meets all outstanding technical requirements.

Kind regards,

Nadeem Nazurally, Ph.D

Academic Editor

PLOS ONE
---

## [Editor Report · Acceptance letter]

29 Oct 2024

PONE-D-24-22940R1 

PLOS ONE

Dear Dr. Curnock, 

I'm pleased to inform you that your manuscript has been deemed suitable for publication in PLOS ONE. Congratulations! Your manuscript is now being handed over to our production team.

Kind regards, 

on behalf of

Dr. Nadeem Nazurally 

Academic Editor

PLOS ONE